# The Dynamics of Indian Labour: Ramifications for Future of Work and Sustainability

**Bino Paul [1,\*], Ramesh C. Datta [1], Unmesh Patnaik [2], Saritha C. Thomankutty [3] and Sumesh P. Soman [1]**

1 School of Management and Labour Studies, Tata Institute of Social Sciences, Mumbai 400088, India; rameshc.datta@gmail.com (R.C.D.); sumesh.soman@tiss.edu (S.P.S.)
2 Centre for Climate Change and Sustainability Studies, School of Habitat Studies, Tata Institute of Social Sciences, Mumbai 400088, India; unmesh.patnaik@tiss.edu
3 School of Vocational Education, Tata Institute of Social Sciences, Mumbai 400088, India; ctsaritha@tiss.edu
\* Correspondence: bino@tiss.edu

**Abstract:** The labour market is complex to systematise into a well-behaved structure. Here, we attempt to understand the synergy between actors and institutions in the Indian labour market and to reveal the implications for the future of work and sustainability. Combining aggregated and micro-level data from multiple sources, we examine the indicators regarding the macro economy, production and engagement in firms, and changes in the occupational structure of workers. While the familiar narrative of technology–labour acrimony emerges, it is weak. In contrast, the tie between economy and technology is a favourable representation of the proximity between human capital and technology. Inadequate human capital implies chances of non-absorption in employment, especially in the future, even for a labour-abundant country such as India. While labour market flexibility through tripartite contact work does not directly relate to labour productivity, high-wage blue-collar work is a more promising factor that aligns with productivity. On the sustainability front, upgrading environmental standards results in more cohesion between labour and capital. The conventional logic of substitutability between capital and labour is invalid when firms adopt environmental standards. From a policy vantage, the scenarios of sustainability transition to cleaner technologies and decent work require complementarity between capital and labour. Considering the dimension of upgrading in order to transition towards sustainability is an essential factor for understanding the future of work.

**Keywords:** future of work; sustainability; capital; labour; environmental standards; India

## 1. Introduction

The labour market is heterogeneous [1–4]. It is too complex to be systematised into a well-behaved structure. Conventionally, it is a system of supply and demand, illustrated by indicators such as labour force participation and number of jobs. While analyses of these indices convey sizes and densities, it is doubtful whether they adequately capture the dynamics. While some changes are endemic to the structure, decisive ones emanate from external sources, especially technology and climate change. Dynamics of this kind have profound implications for the future. Changes in the labour market involve complex interactions between institutions and interests that the conventionally accepted frames and indices may fail to capture. We attempt to understand these synergies by examining the future of work and sustainability in the Indian labour market. In particular, the paper's contribution attempts to unravel answers to questions such as: "Is there a triadic relationship between the labour market, technology, and aggregate economy?", "How is human capital likely to influence the future of work scenarios?", "Does labour market flexibility enhance productivity in the economy?", and "What role do all of these factors play in the transition towards sustainability?".

The past two decades have been a landmark concerning the evolution of the future of work. Radical changes have emerged in how we work due to sources such as artificial-

intelligence (AI)-based technologies or the occurrence of a global pandemic. These events are dynamic, impacting human engagement in the labour markets. However, their impacts may vary; for some people, it is displacement from occupations such as unemployment or non-participation in the labour market [5]. Nevertheless, these technologies may be complementary to some occupations. Another crucial aspect is economic efficiency. Technological change may alter the profit margin by reducing the value/supply chain cost and labour. AI-based systems such as platform economies use data and analytics extensively to recommend customer choices [6]. On the final demand side, it culminates in lower prices. This means that last-mile delivery is at the lowest price.

The future of work consists of giving and taking. If the count of losers substantially exceeds gainers, it paves the way for inequalities. If the labour share in value plunges to a critical low, consumption and saving suffer. It could even lead to clusters of homogenous labour alienated from employment relations. Assets accumulate in the hands of a few because of soaring profits, while the labour share drops. For many, after a long separation from employment, they end up joining a hapless disengaged class. In such a scenario, future of work is tilted towards a disequilibrium with polarised benefits symbolised by a few highly wealthy people coexisting with a massive stock of homogenous precarious classes. However, the path towards rebalance could emerge with newer variants of collective bargaining deterring the balance of those who have versus those who do not from becoming self-sustaining. Not only do tangible outcomes such as income inequality occur due to the future of work, but it (due to AI) also changes employment contracts. It brings a control regime, replacing the technical and bureaucratic frame of direction, evaluation, and discipline. Instead, it there will be a new control system driven by AI and data.

The multidimensional sustainability angle covers environmental, economic, and social dimensions [7,8]. In the environmental domain, the core issue is the relationship between technology and the use of resources. For example, a polluting, but cheaper technology will generate effluents that render a downstream water body unusable. It will create a spiral of adverse outcomes. Upgrading to a cleaner technology would mitigate this effect. The economic aspect is about costs and benefits, both tangible or intangible. If costs overrun benefits, an economic unit is destined for closure. At times, the scope of environmental and economic elements may concur, while it may sometimes diverge.

The social facet is more intricate than the others [9,10]. It deals with the social–psychological–economic well-being of people and is more idiosyncratic. For example, a global value chain that uses supplies from a source of vulnerable employment will have diminished prospects in international business. It calls for social upgrading the value chain. Another social sustainability obstacle is the persistence of precarity in employment [11,12]. Linking the sustainability angle with the labour market condition is crucial for the future of work. If the scenario is reduced employment, it is challenging to transform displaced workers with skills for new occupations. It requires a sustainable stream of investment that earns a social return on investment. Investment of this sort warrants a need for social entrepreneurship and social innovation. It calls for new institutions and collaborations. Another crucial scenario is a circular economy [13]. The conventional economic system generates value and externalities such as solid or liquid waste. If the waste is recycled or reused, it contributes to environmental sustainability. Alternatively, it also saves resources for posterity. The future of work in such an extreme is transitioning from a linear economy to a circular economy, thus creating green jobs.

## 2. Data and Methods

The Indian labour market has presented a number of issues, such as a gigantic precarious and informal workforce, a tenacious and perceptibly lower female workforce participation rate, burgeoning youth unemployment, a plummeting labour share in the gross domestic product, and a stagnant real wage in manufacturing. We used four datasets for this analysis. Penn World Data examines the relationship between macroeconomic aggregates [14]. We extracted the data for India for the period of 2000 to 2019. Essentially, these

variables depict the traditional factors of production and their contribution to output, skills, and productive efficiency. Table S1 provides the list of variables and their definitions.

Employment status is measured as the number of persons employed. In the dataset, the economy's gross domestic product is adjusted for inflation as it is measured at constant prices, with the year 2017 being the base. While the measurement of capital stock uses 2017 prices, work hours refer to the average working hours devoted by the persons employed. Labour productivity and the capital–labour ratio show the contribution of labour and capital to the economy's total output. Human capital reflects the contribution of health and education to the productive capacity of workers. Total factor productivity, otherwise known as multifactor productivity, shows the ratio of aggregate outputs to aggregate inputs. All of the above variables are used in log-transformed form for the analysis in order to reduce skewness in the data (the descriptive statistics are presented in the Supplementary File). The analysis of indicators is for the time frame of 2000–2019 at an all-India level.

The second dataset is the Periodic Labour Force Survey 2019–2020, published in July 2021 [15]. The field survey was conducted from July 2019 to June 2020, and the sample size consisted of 0.10 million households at all India levels. Within this sample, approximately 0.24 million and 0.18 workers participated in the survey across rural and urban regions, respectively. PLFS presents data on critical employment and unemployment indicators, along with the activity status of people. It presents the activities pursued by a person during the reference period, that is, the last 365 days preceding the survey date, also called the usual activity status of the person. We analysed the microdata to examine the dynamics of occupation. A third dataset was used for temporal comparison of the employment trends. This was the National Sample Survey 68th round of microdata on India's employment and unemployment situation [16]. The field survey from data collection for this round was from July 2011 to June 2012, and the data were released in January 2015. The total sample size of the survey was just over 0.1 million households and included approximately 0.45 million households. This sample included households from both urban and rural areas. In rural areas, 0.28 million workers participated in the surveys, while in urban areas, the sample size was 0.17 million.

The fourth dataset was microdata from the Annual Survey of Industries for 2017 to 2018, released in June 2019 [17]. It is one of India's principal sources of information on industrial activity. It presents firm-level data on output, raw material, capital and labour, and factor payments, as well as characteristics of the firm, such as ownership and nature of enterprise, and it is updated annually. The reference period for this survey was the financial year, stretching from April to March of a particular year and covering all factories in the country, as defined by the Indian Factories Act 1948. This survey complied with the legal mandate of The Collection of Statistics Act (2008), the revised version of The Collection of Statistics Act, 1953. The sample size for this round was 76,613 units spread across India. Table S2 presents the variables used in the analysis along with their definitions.

We used multiple analytical methods to examine firm and labour market behaviour dynamics [5]. This included trend analysis, bivariate plots, the network of variables, and random forest classification. The trend analysis showed the temporal evolution conducted when selecting variables characterising the macroeconomic structure of the industry and the labour market. The bivariate plots aimed to capture the empirical relationship between two variables. The overall objective of such an analysis was to visualise the pattern and degree of relationship between the underlying variables. However, they failed to capture variegated relationships between several variables. Therefore, we used network analysis to understand the structure and function of complex systems created by the interaction of many variables in the economy and industry. The final method used was the random forest classification [18,19], which explored the sustainability angle. It is a machine-learning algorithm that aggregates the output from numerous decision trees. We considered environmental management as a decisive variable and tried to understand the convergence between labour and capital.

## 3. Results and Discussion

### 3.1. The Evolution of Labour, Technology, and Economy at an Aggregate Level

The temporal trend of the three dimensions, labour, economy, and technology, are presented in Figure 1. It is a panel of six graphs plotting the trend over nearly two decades, spanning from 2000 to 2019. The dimension of labour includes employment (panel A), average hours of work (panel E), and labour share (panel F). Gross domestic product (panel B) represents the economy. Capital (panel C) and capital services (panel D) form the technology. For economy and technology, the trajectory is relatively linear. It does not deviate from the plotted smooth curve. However, labour is more non-linear. Concerning labour share, it plummets over the first decade and then shows a slow move in an upward direction. The fall for this variable is too steep to recover. Employment shows a non-linear trajectory. While the first decade shows a steeper expansion, the second decade reports a flatter curve. This implies that India is headed towards a scenario of economic expansion, bringing hardly any jobs. Alternatively, this could be jobless growth resulting from increased work hours. However, it is approaching a plateau. The crux of the matter is that labour dynamics need to be in sync with technology and the economy. Figure 1 provides scope for further explanations. It is crucial to look into technology, productivity, and human capital. Figure S1 (see Supplementary Materials) conveys the trends of capital per labour (panel A), labour productivity (panel B), human capital (panel C), and total factor productivity (panel D). An interesting pattern is that total factor productivity shows higher growth post-2010. Human capital and labour productivity show a consistent linear trend for growth. Capital per labour shows steep growth during the first decade. However, its trajectory is linear during the second decade. Labour productivity and total factor productivity are crucial outcomes in an economy. Moreover, capital per labour and human capital are consistently increasing. Nevertheless, the essence of the results is that the labour trend needs to conform to the consistent growth in technology and the economy.

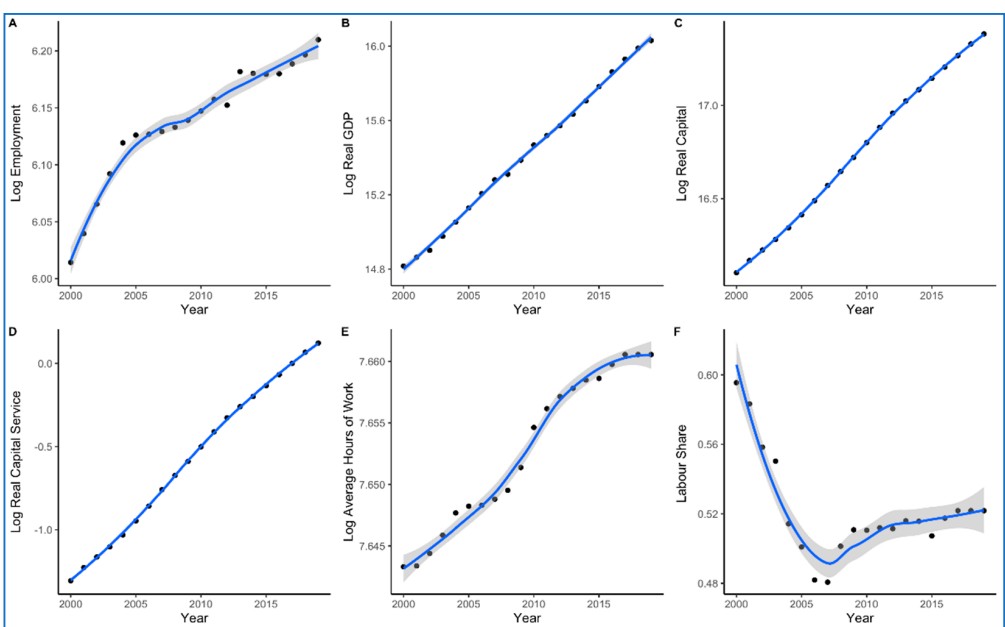

**Figure 1.** Labour, economy, and technology in India during 2000–2019. (**A**) shows the behaviour of employment, (**B**) depicts real GDP, (**C**) captures real capital, (**D**) captures capital per employed, (**E**) shows human capital and (**F**) refers to labour share; The dots show the specific value across the years, while the blue line depicts the trend (Source: Author(s) computation from Penn World Data 10.10 [14]).

Bivariate graphs show that the interaction between the economy, labour, and technology is crucial. Figure 2 presents a panel of six graphs. It includes pairs of GDP and employment (panel A), capital and employment (panel B), hours of work and employment (panel C),

labour share and capital per employed (panel D), total factor productivity and human capital (panel E), and labour share and labour productivity (panel F). Panel A depicts a non-linear pattern. For higher values of GDP, signifying the second decade, its expansion corresponds to a much lesser employment increase than during the earlier phase. The same behaviour validates the relationship between capital and employment (panel B). It implies that a steeper change in the capital is aligned with a sluggish increase in employment. For most of the period, a slight change in employment corresponds to a spike in working hours (panel C). It results in a plateau. For the lower range of capital per employed, during the first decade, labour share declines steeply. Then, it rises gradually (panel D). Graph E shows the relationship between total factor productivity (TFP) and human capital. TFP accounts for the variation in value-added that emanates from sources other than labour and capital. The variable human capital is based on years of schooling and returns to education. The covariation between TFP and human capital is becoming stronger. Skills and higher educational attainment complement economic growth driven by technology and knowledge. Graph F shows that the relationship between labour share and productivity is non-linear. It is a steep inverse pattern for the lower values corresponding to the first decade. However, it gradually settles at a slighter higher value during the second decade. This implies that labour productivity and labour share diverge substantially. The former is in the group of economy and technology, while the latter is with labour.

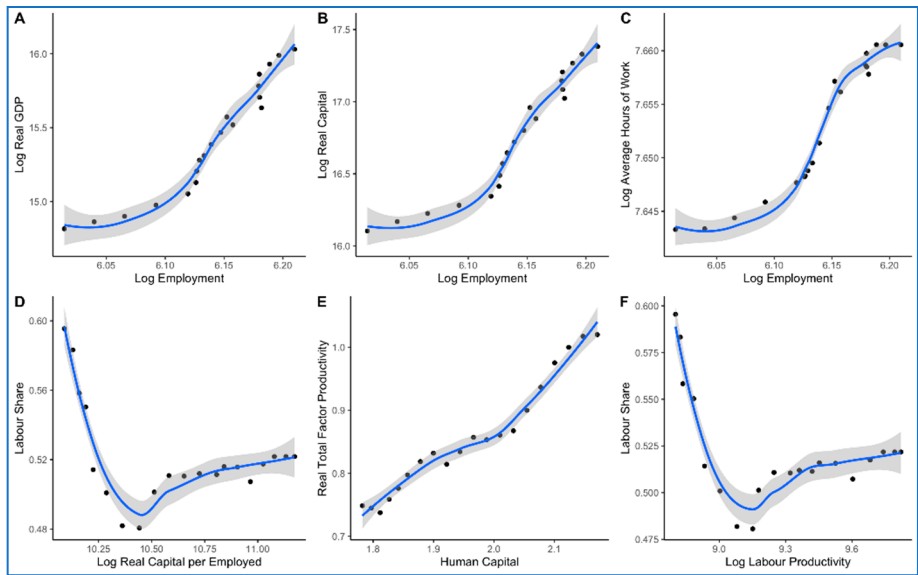

**Figure 2.** Bivariate plots between the economy, employment, human capital, and labour productivity. (**A**) depicts the relationship between GDP and employment, (**B**) is between capital and employment, (**C**) captures the relationship between average working hours and employment, (**D**) shows bivariate relationship between labour chare and capital per employed, (**E**) depicts the relationship between productivity and human capital, (**F**) depicts the relationship between labour share and productivity. The dots show the specific value across the years, while the blue line depicts the trend (Source: Author(s) computation from Penn World Data 10.10 [14]).

The above analysis, for the trends and relationships, shows the aggregate structure. Labour stands alone, as evident in the case of employment and labour share, while the economy and technology are in sync with each other. We visualise two networks to capture the nature of the interaction between labour, economy, and technology (Figure 3). The network analysis of variables is distinct from multivariate tools such as regression. First, there is no independent variable—every variable is correlated with other variables. Second, the correlation is the base for weighing the mutuality of variables. Using the EBICglasso estimation, a network is generated. It is a valued undirected network. Furthermore, the estimates are non-parametric.

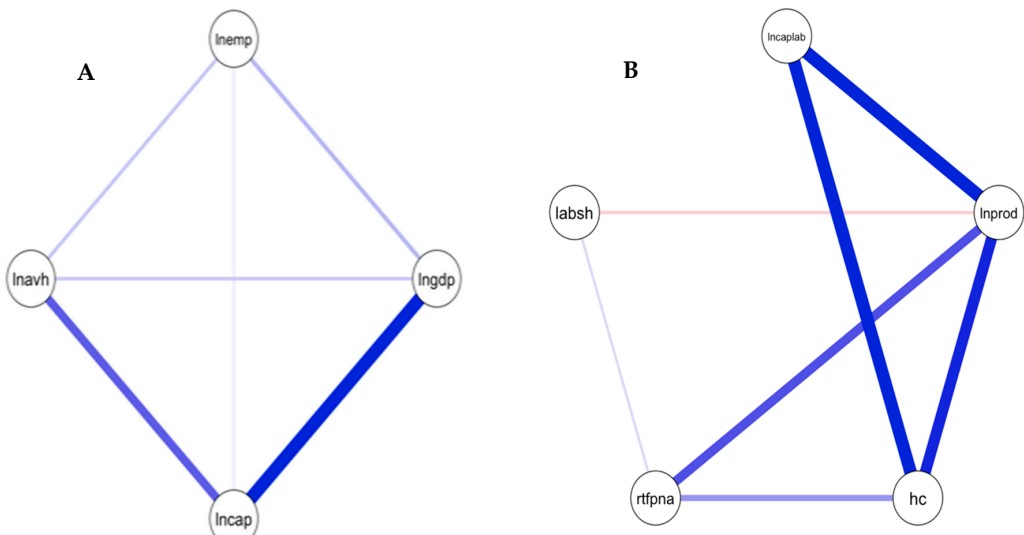

**Figure 3.** Network between labour, capital, GDP, and human capital during 2000–2019. (**A**) presents the connection between employment, average hours of work, GDP and capital. (**B**) presents the links between labour share, labour productivity, human capital, capital per labour and total factor productivity. The lines in the diagram are bi-directional ties. Thickness of the line depicts the strength of the tie. Blue indicates a positive tie and red signifies a negative tie (Source: Author(s) computation from Penn World Data 10.10 [14]).

Network A presents the connection between employment, average hours of work, GDP, and capital. Labour is represented by two variables, namely employment and average work hours of the employed, and gross domestic product captures the economy. The proxy for technology is real capital. The tie between GDP and capital is the strongest (Table S4), implying an intense proximity between economy and technology. In contrast, the tie between employment and capital is the weakest. The pattern points to the tension between labour and technology. However, a strong relationship exists between average work hours and technology. This means that more technological or capital deepening accompanies more work hours. Therefore, the network points to the close ties between technology and the economy, while labour is not part of it. It is crucial to note that GDP growth and employment expansion is far below equiproportional.

Network B consists of five variables: labour share, labour productivity, human capital, capital per labour, and total factor productivity. The network's strongest tie is between productivity of capital and human capital (Table S4). In contrast, the weakest tie is between total factor productivity and labour share. Interestingly, capital productivity, labour productivity, and human capital form a strong triadic alliance. This is a case of productivity, technology, and human capital acting coherently. Another triad consists of productivity, human capital, and total factor productivity. In this formation, productivity and human capital form the strongest tie. A crucial finding is that the tie between labour share and productivity is negative. It is apparent from network B that labour needs to be in sync with the relational structure of technology, productivity, and human capital.

There is a disentanglement between labour and the duo of economy and technology (Figure 3). It has clear implications for the future of work. The tie between economy and technology appears to correlate with human capital. However, the duo is either weakly connected with employment or is adversely related to the labour share. Another consideration is the need for a direct engagement between productivity and labour share. In a nutshell, there needs to be more job creation in the economy. In contrast with this, there is a visible deepening of technology. Both labour productivity and total productivity show a discernible increase. However, these forces favour the labour with more human capital while excluding those with lesser human capital. Assessing the labour market outcomes

of human capital, particularly the occupational structure, is crucial. The following section looks into the dynamics of the occupational structure in India.

### 3.2. Occupation Dynamics in the Indian Labour Market

The employed in the labour market belongs to diverse occupations. In India, the National Classification of Occupation (NCO) defines occupations (Table S5). It consists of three levels: (a) single digit, (b) two-digit, and (c) three digit. While the single-digit classification is the broadest, three digit has the most granular data. The single-digit classification breaks the employed into nine broader categories: legislators, senior officials and managers, professionals, technicians and associate professionals, clerks, service workers and shop and market sales workers, skilled agricultural and fishery workers, craft and related trade workers, plant and machine operators and assemblers, and elementary occupations. Figure 4 depicts the size of occupational categories in 2011–2012 and 2019–2020. Irrespective of years, the occupation order regarding the size remained the same, while all categories grew during this period, except for the "elementary occupations" category. The category "skilled agricultural and fishery workers" reports the highest (156 million in 2019–2020). It consists of market-oriented skilled workers and subsistence workers. The second-highest category is "elementary occupation" (113 million in 2019–2020). It includes sales and elementary service occupations, agriculture, fishery and related labour, mining, construction, manufacturing and transport. Interestingly, during the analysis, this occupation contracted by nearly 10 million. Intuitively, in 2019–2020, Legislators, Senior Officials and Managers, Professionals, Technicians and Associate Professionals, and Clerks constituted the 'White Collar' labour, aggregating 94 million, while the rest of the categories, representing the 'blue collar' labour, sums to 398 million.

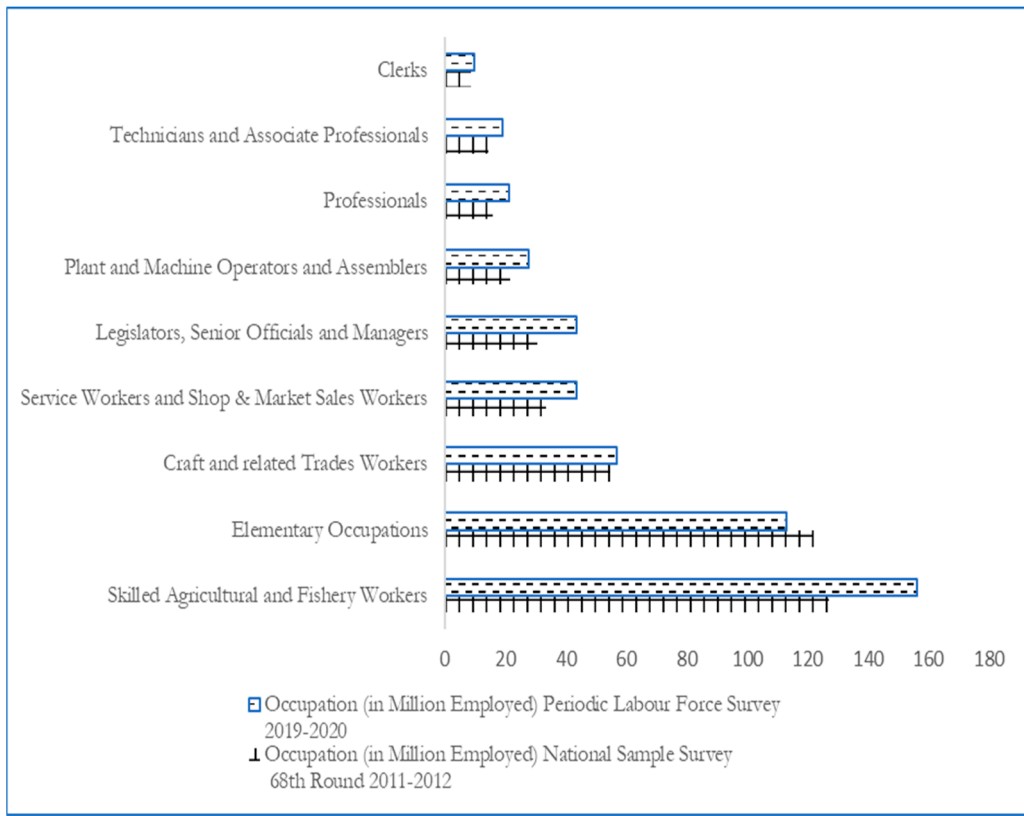

**Figure 4.** Occupations (in Million) during 2011–2012 and 2019–2020 (Source: Author(s) computation from unit records of NSS 68th Round [15] and Periodic Labour Force Survey 2019–2020 [16]).

Another way of distinguishing occupational categories is to examine employment status distribution within occupations (Figure S2). Employment status consists of three

types: self-employed, regular, and casual. While neither employee nor employer is employed, the other two involve these roles. In the case of those regularly employed, employment relation exists formally or informally, generating a durable engagement and assured pay. However, employment is uncertain for the category "casual"; there is work today, but not necessarily tomorrow. Self-employment is the significant type in two categories: legislators, senior officials, and managers, and skilled agricultural and fishery workers. For other white-collar occupations, regular employment is discernibly the principal category. Concerning elementary occupations, casual employment constitutes the most significant chunk. However, employment status is mixed with other occupational types.

An interesting phenomenon during the analysis period is the discernible variation in the growth of occupations. The growth rate varies across occupational streams (Figure S3). Except for "clerks", other white-collar categories report visibly higher growth, while "clerks" show significantly lower growth. Compared with the phenomenal growth of white-collar employment, blue-collar labour shows moderate to lower growth. A notable case is "elementary occupations", showing the lowest growth. Moreover, approximating these patterns, the higher the volume of the labour force in an occupation, the lower the magnitude of the growth rate. Institutively, more expansion in the labour force tends to generate lower growth in wages (Figure S4). An exciting pattern emanates from the analysis of the wage differential (WD). It refers to the ratio of the difference between the median wage of another occupation and any particular occupation to the median wage of that particular occupation. For any particular occupation, there will be eight WDs. The series of these WDs generates an average WD. As depicted in Figure 5, while white-collar labour shows the highest average WD over other occupations, it is substantially lower for blue-collar labour. Nevertheless, for white-collar work, it drops considerably during the analysis period.

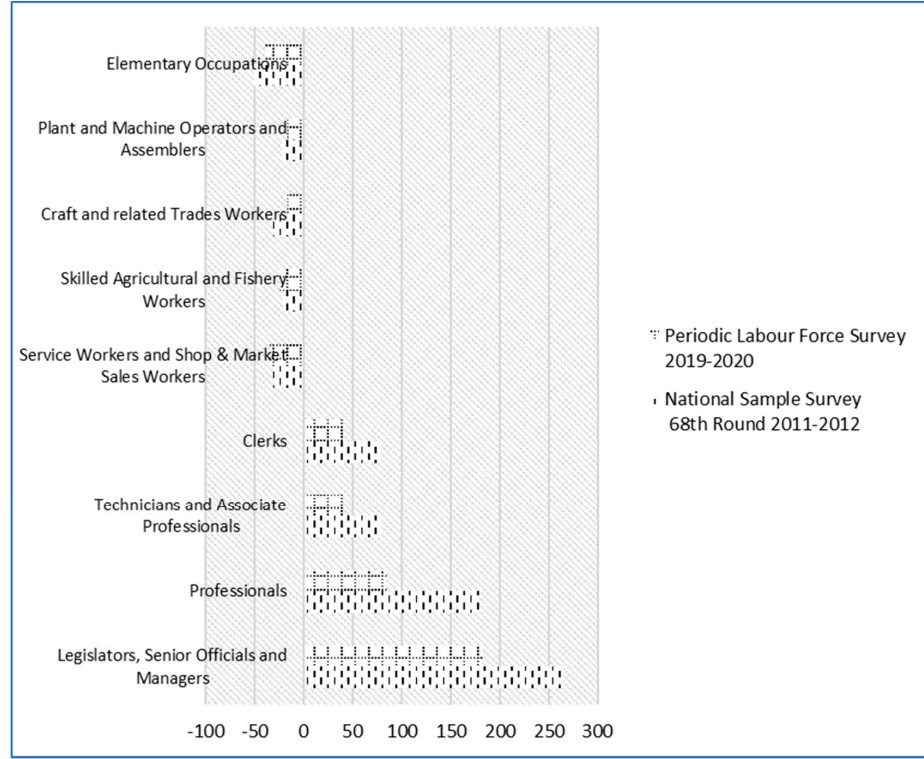

**Figure 5.** Average wage differential between any occupation with other occupations during 2011–2012 and 2019–2020 (Source: Author(s) computation from unit records of the NSS 68th Round [15] and Periodic Labour Force Survey 2019–2020 [16]).

### 3.3. Labour, Technology, and Value Added: The Dynamics at Factories

Previous sections examined macroaggregates and the dynamics of the occupational structure. These bring out polarising scenarios in the labour market. The analysis of the macroaggregates reveals that the coherence between labour and the dyad of technology and economy could be more robust. It is clear that economic growth and technological deepening scarcely reflect the expansion in employment and share of labour in national income. Moreover, labour must acquire human capital to participate in a technology-led economy. The household data unravel the dynamism of the occupational structure. There is a visible contraction in cheap wage segments, while white-collar occupation shows considerable growth despite its small share in the labour force. However, the average wage differential the white-collar group (mainly managerial and professional streams) enjoyed declined during the years 2011–2012 and 2019–2020. It is interesting to place these findings together. Employment in India is gradually tilting towards supervisory and managerial pursuits, while it is losing momentum for low-wage occupations in primary and secondary activities. Another stream of data that helps our analysis is Annual Survey Industries (ASI) plant-level data covering manufacturing and allied activities. We rely on the microdata for the year 2017–2018. There are four relational structures (networks) and thirteen variables. Some variables are unique to a particular network. However, some have two affiliations (Figure 6 and Table S6).

Network A consists of value added, capital, and labour. The other two variables are also part of network B, except for labour. Instead, three different types of employees are part of the network covering manager, directly employed workers, and contract workers. Concerning network C, there are four variables: productivity, capital per employed, the share of supervisory and managerial staff in employed, and the share of contract workers. Network D consists of productivity, capital per employed, wage rate of supervisory and managerial staff, wage rate of directly employed workers, and wage rate of contract workers. Table S7 provides the descriptive statistics of these variables. The relation between pairs of value added and labour is the strongest. Moreover, a moderate positive tie between value added and capital. However, the relationship between capital and labour is fragile. This means that labour is very crucial for business performance. Technology also contributes to performance. However, technology and labour are not in sync with each other. However, this inference is sensitive to the density of labour. For higher values in the scale of capital, the pair of capital and labour demonstrates a higher degree of unidirectionality. It implies that the tension between capital and labour is heterogeneous across capital intensity.

Concerning network B (Figure 6), it retains value added and capital. However, labour has three segments: count of mangers, direct workers, and contract workers. Therefore, the network has five variables and ten pairs of relationships. The strongest tie among the pairs is between directly employed workers and value added. Furthermore, the weakest positive tie is between capital and contract labour. Although supervisory and managerial staff (white collar) share moderate positive ties with directly employed workers and contract workers, the tie between both categories of workers among the blue collar is a weak negative one. Another exciting pattern is the weak tie between capital and any employee. Among the employed, the white-collar segment reports the highest weight. Value added and capital density and correlation plots show relatively more symmetric distributions. However, density plots look more asymmetric concerning the white and blue collar. Any pair involving value added exhibits a linear fit without being impactfully pulled by the outliers. However, lower range values show a scattered pattern concerning capital, while the upper range is unidirectional. Network C provides a relational structure of productivity, capital per employee, and proportion of the white collar and the share of contract workers. The most substantial tie exists between capital per employee and productivity.

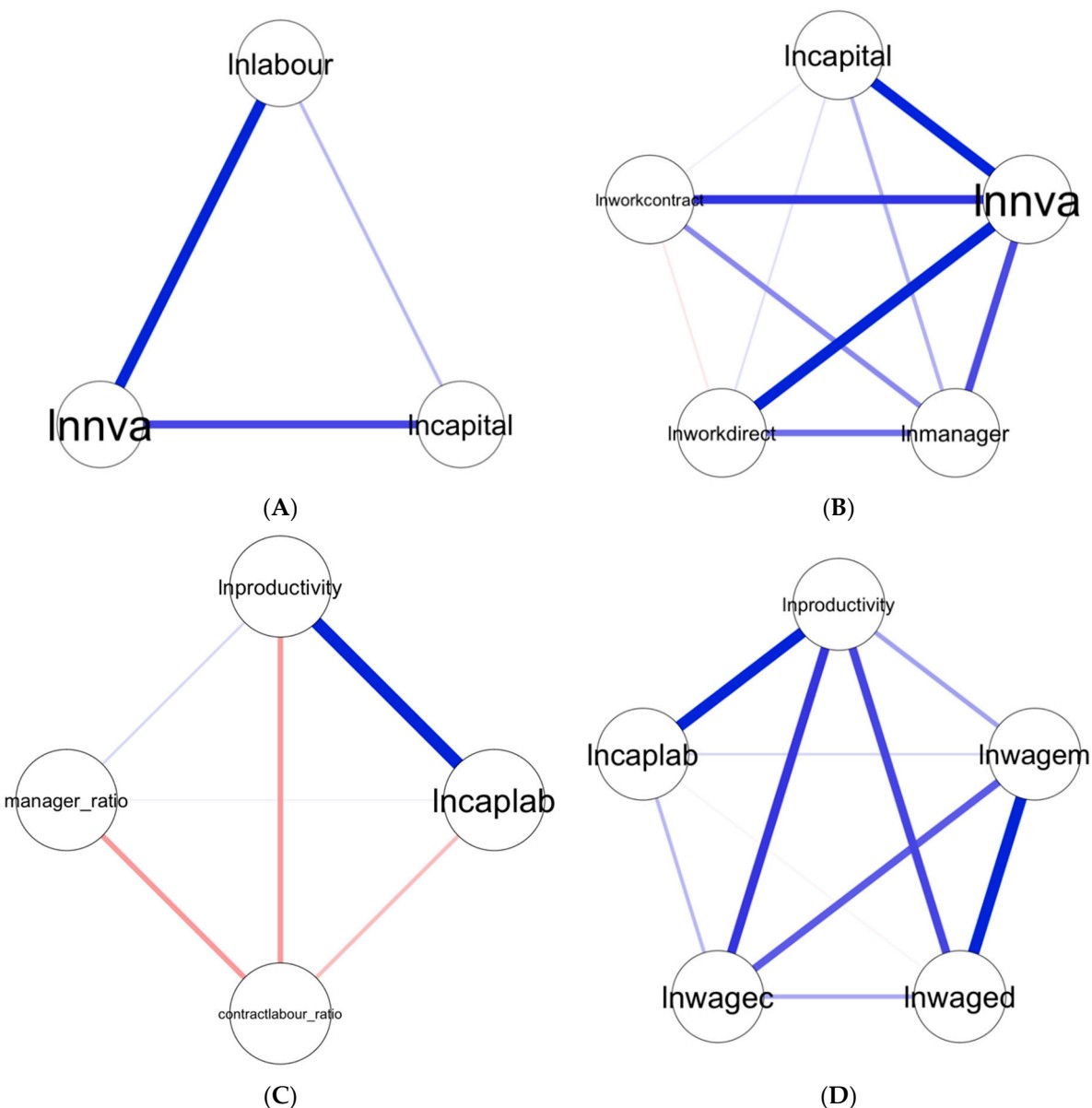

**Figure 6.** Network of variables between labour, technology, and value added. (**A**) depicts the network links between labour, capital and value added, (**B**) shows the formation of network between value added, capital, managers, directly employed and contract workers, (**C**) shows the network involving productivity, capital per employed, share of supervisory and managerial staff in employed and contract workers, (**D**) presents network links between productivity, capital per employed and wage rate of types of workers. The lines in the diagram are bi-directional ties. The thickness of the line depicts the strength of the tie. Blue indicates a positive tie and red signifies a negative tie (Source: Author(s) computation from the unit records of the Annual Survey of Industries (2017–2018) [17]).

A striking result is the weak negative tie between productivity and the share of contract workers. On the other hand, the white-collar segment and productivity pair show a weak positive relation. However, the relationship between the share of contract workers and productivity could be more vivid and distinct from a scattered pattern. Moreover, it is valid for any pair other than productivity and capital employed. The result conveys that labour market flexibility founded on tri-patriate contacts may have limitations in generating productivity gains. If the aggregate economy relies on productivity as a driver, the informal tripartite contract employment system needs a refresh regarding skills and entitlements. Furthermore, plots of density are a mix of symmetric and asymmetric shapes.

There are five variables in network D, consisting of productivity, capital per labour, wage rate of supervisory and managerial staff, wage rate of directly employed workers, and wage rate of contract worker. Wages rates for blue collar show a strong positive relationship with productivity. However, the wage rate of white collar is weakly related to productivity. Another remarkable result is the weak pairing between technology, represented by capital per labour, and the labour for white or blue collar. The correlational plots show that the relationship between wage rates and technology does not show a clear pattern. However, concerning white collar, the higher range of wage rate directly varies with technology.

### 3.4. The Sustainability Dimension

The acrimony between capital and labour has been a theme of intellectual curiosity for years. It is a divided camp. Views and evidence vouch for its convergence. However, the opposing views also have their facts. This raises a question. Is the acrimony between labour and capital valid across contexts? This question is crucial from the angle of social and economic sustainability. Disengagement between capital and labour manifestly reflects plummeting labour share, potentially contributing to increasing socio-economic inequality. It may have ripple effects on the aggregate demand. As a result, businesses and livelihoods may tend to become crippled. Is there no hope? We examined whether environmental standards bring a different depiction. This analysis relied on a variable that coveys whether a factory attained International Standard Organisation (ISO) 14000 [20], and is focussed on environmental management.

Only one-tenth of factories in the sample attained sustainable environmental management practices. A crucial question is if the relationship between value added, capital, and labour is sensitive to achieving the ISO. Furthermore, does the disengagement between capital and labour vary with the attainment of ISO? Figure S5 reveals the correlation plots and density for both the attainers and non-attainers groups. There is a visible difference between these groups in their correlations and densities. Concerning attainers, the plots of density seem symmetric. In contrast, plots for the other group are a mix of symmetric and asymmetric distributions. The most striking difference between these two groups is that attainers show a coherent linear, unidirectional relationship between capital and labour. However, the pattern for non-attainers is a mix of scattered points and directly covarying points. Is this visual story aided by noises emanating from unobserved heterogeneities? This is a crucial question. We used random forest classification (a machine learning algorithm) to assess the veracity of these patterns. The algorithm splits data into three sets: training (64%), validation (16%), and testing (20%) (Supplementary Materials S1). The test set is akin to a randomised trial. As shown in Figure 7, the attainers lie clustered across plots, distinct from non-attainers. The difference between attainers and non-attainers is far from an outcome of serendipity. The above result is a silver lining. A firm's upgrading may go along with performance, resources, and sustainability. Perhaps it is crucial to envisage a convergence of social, economic, and environmental sustainability.

The above inference is not merely regarding the complementarity between capital and labour. Instead, it presents the scenario of upgrading the value creation. A firm that internalises the logic of environmental, social, and economic sustainability relies for its growth for upgrading to cleaner technologies and developing a skilled and decent workforce. While ISO 14000 is a proxy for process improvement or innovation, it is a prerequisite to the upgrading efforts of a firm [21]. More succinctly, upgrading also involves the scope for a sustainability transition. A firm that relies on cheap labour as a comparative advantage may abstain from investing in cleaner technologies. However, a strategy of this kind brings sustainability risk to the governance of the value chain. Therefore, the firm must elevate itself from the conventional conundrum of capital and labour acrimony to symbiotic possibilities.

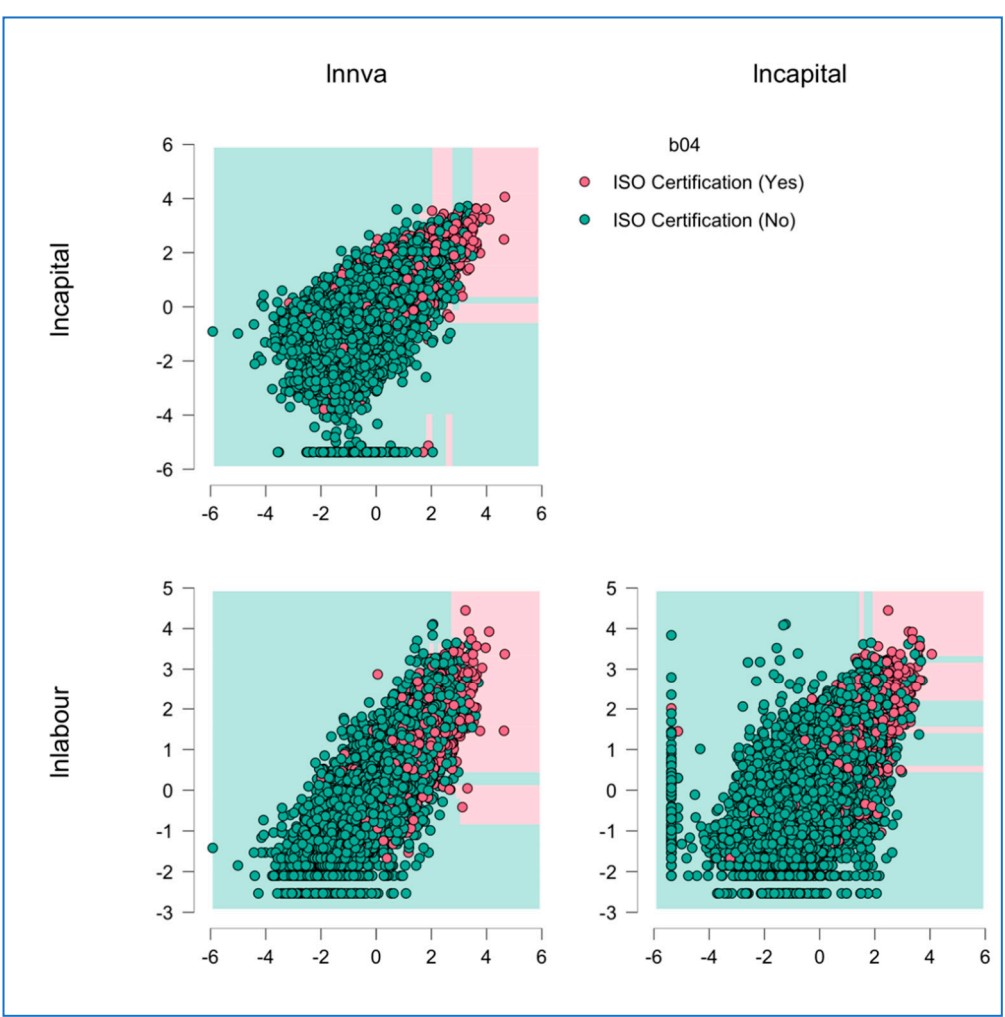

**Figure 7.** Random forest classification (Source: Author(s) computation from the unit records of the Annual Survey of Industries (2017–2018) [17]).

## 4. Conclusions

The analysis shows a weak tie between capital, a good proxy for technology, and labour [22]. Furthermore, it resonates with wages and working hours. In contrast, the tie between economy and technology is an example of bonhomie. Both dimensions are aligned, be it a time series or a cross-sectional context. Another crucial thing is the proximity between human capital and technology. A technology-induced job requires human capital. Inadequate human capital implies chances of non-absorption in employment, especially in the future. Even a labour-abundant country such as India is not an exception to this possibility. An exciting finding is that labour market flexibility through tripartite contact work does not directly accompany labour productivity. However, high-wage blue-collar work may be a more promising factor that aligns with productivity. Upgrading the standards, especially environmental ones, brings capital and labour together [23]. Although the extant views posit capital and labour as substitutes, they may need to be validated in the context of sustainability transition [21]. Here, a case in point is that the firm that opts for better environmental standards also tends to see the complementarity between labour and capital [24]. Bringing cleaner technology is more than just a question of factor substitution. Instead, it involves upgrading knowledge and human resources for better outcomes. From a strategic point of view, the symbiosis between labour and capital is crucial for a sustainability transition that assures cleaner production and decent work, especially in the future of work. While this study is specific to an Indian context, patterns and inference may have relevance for other emerging economies.

**Supplementary Materials:** The following supporting information can be downloaded at: https://www.mdpi.com/article/10.3390/su15129312/s1.

**Author Contributions:** Conceptualization, R.C.D.; Methodology, B.P., R.C.D., U.P. and S.P.S.; Validation, R.C.D. and S.C.T.; Formal analysis, B.P., U.P., S.C.T. and S.P.S.; Data curation, B.P. and S.P.S.; Writing—original draft, B.P., R.C.D. and U.P.; Writing—review & editing, U.P., S.C.T. and S.P.S.; Visualization, B.P., S.C.T. and S.P.S. All authors have read and agreed to the published version of the manuscript.

**Funding:** This research received no external funding.

**Institutional Review Board Statement:** Not applicable.

**Informed Consent Statement:** Not applicable.

**Data Availability Statement:** The data can be made available to researchers on reasonable request to the corresponding author.

**Conflicts of Interest:** The authors declare no conflict of interest.

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
