# Peer review of "The Dynamics of Indian Labour: Ramifications for Future of Work and Sustainability"

_sustainability, doi:10.3390/su15129312_

Round 1
Reviewer 1 Report
Overall this is an informative and thought-provoking paper looking at relationships among Indian labor , capital and technology in the context of sustainability and environmental factors. The paper actually suggests that the relationship between capital and labor is complementary in the context of sustainability and environment rather than as substitutes.
The authors have developed four data sets upon which they based their analysis. These data sets are a strength of the paper.
The only real question that I would raise is a decision for the editors. Are these findings generalizable or do they apply only to India? I found it interesting both from a methodological and substantive theory standpoint but I’m not sure that other readers would feel the same way. I have made a few specific comments below.
Abstract:
Quite clear and informative.
Introduction:
Page 2 line 54 uses the phrase for some people it may be displacement from occupations. Does this mean unemployment?
I do not know what the journal’s policy is on multiple citations. A number of journals that I review for do not like them.
Data and methods:
Page 3 line 117 - all data have been log transformed to deal with hetergeneity and skewness of the data. It might help the reader to know the extent of the heterogeneity and skewness.
Results and discussion:
Page 4 line paragraph beginning at line 162 could use a minor edit. For example, I am not certain that “curvier” is a word or whether the phrase ‘more curved’ would be appropriate. “It hovers around a slightly higher up”seems pretty awkward.
The change in figure discussion from Figure 1 to Figure S1 is a bit confusing. Are the supplementary tables going to be published along with the paper? It would seem to be necessary given the discussion. I can see that this might provide a paper length problem.
The discussions here are qualitative in nature. The paper could gain scientifically by doing correlation analysis or perhaps some simple regressions.
Page 5 first paragraph - same comment about the need for editing.
The authors might present the slopes of the graphs in figure 2 to add to their qualitative statements about two segments for the graphs.
The network analysis is a strength of the paper.
Conclusion:
Very well stated and very much consistent with the results.
See comments to authors
Author Response
Comment: Page 2 line 54 uses the phrase for some people it may be displacement from occupations. Does this mean unemployment?
Response: Yes, it is either unemployment or non-participation in labour market.
Comment: Page 3 line 117 - all data have been log transformed to deal with hetergeneity and skewness of the data. It might help the reader to know the extent of the heterogeneity and skewness.
Response: The log transformation is done primarily to address skewness. The descriptive statistics are presented in the supplementary file in Table S3 and Figure S5.
Comment: Page 4 line paragraph beginning at line 162 could use a minor edit. For example, I am not certain that “curvier” is a word or whether the phrase ‘more curved’ would be appropriate. “It hovers around a slightly higher up”seems pretty awkward.
Response: Corrected in the revised version.
Comment: The change in figure discussion from Figure 1 to Figure S1 is a bit confusing. Are the supplementary tables going to be published along with the paper? It would seem to be necessary given the discussion. I can see that this might provide a paper length problem.
Response: The supplementary materials will be available online along with the manuscript.
Comment: The discussions here are qualitative in nature. The paper could gain scientifically by doing correlation analysis or perhaps some simple regressions.
Response: Tables S4 and S7 provide network weights that are based on paired corelations. Figure S5 provides correlational plots and density of variables. The discussions draw cues from the pattern emerging from networks and correlational plots. Further it is supplemented by the implications.
Comment: Page 5 first paragraph - same comment about the need for editing.
Response: It is addressed in the revised version.
Comment: The authors might present the slopes of the graphs in figure 2 to add to their qualitative statements about two segments for the graphs.
Response: The bivariate plot illustrated in figure 2 is estimated within a 95 percent confidence interval. Therefore it is an outcome of an inferential process. Furthermore the slopes tend to vary due to changes in policy context.
Comment: The network analysis is a strength of the paper.
Response: We thank the reviewer.

Reviewer 2 Report
Something was fundamentally missing in this article including rationale and hypotheses. There was no clear statement that gave me any direction on the data gathering.
It was OK.
Author Response
Comment: Something was fundamentally missing in this article including rationale and hypotheses. There was no clear statement that gave me any direction on the data gathering.
Response: The data set involves both secondary and microdata sources. The supplementary section provides the definition of the variables, and sources of data. Section 2 in the manuscript elucidates the data and methods. It discusses details of the sources, covering Penn World Data, Periodic Labour Force data and Annual Survey of Industries data. The rationale is made more explicit in the revised version (Introduction section).

Reviewer 3 Report
Comments
The study addresses a topic of interest and current affairs. The author draws attention to three complex interactions work, economy and technology, as well as the factors that influence different processes and scenarios. On the one hand, the importance of a sustainable balance (homeostasis - resilience) between the environment. Work and technology. In this regard, we must not stop considering that prior to the pandemic, new technologies were transforming jobs. The pandemic imposed a greater speed on this change. The problem is that our labor laws and the new forms of work (AI) are not aligned. This leads to establishing new occupational health problems that are not considered in this study since the analysis puts the point on more economic than social issues.
Author Response
Comment: The study addresses a topic of interest and current affairs. The author draws attention to three complex interactions work, economy and technology, as well as the factors that influence different processes and scenarios. On the one hand, the importance of a sustainable balance (homeostasis - resilience) between the environment. Work and technology. In this regard, we must not stop considering that prior to the pandemic, new technologies were transforming jobs. The pandemic imposed a greater speed on this change. The problem is that our labor laws and the new forms of work (AI) are not aligned. This leads to establishing new occupational health problems that are not considered in this study since the analysis puts the point on more economic than social issues.
Response: It is beyond the scope of the paper.

Reviewer 4 Report
In introductory part and even the initial part of the method, descriptions and statements are made that require theoretical or empirical support or perhaps a statistical source that supports some statements, for example, the behavior of the market, the economy, employment, etc.
In general, it seems to me that greater theoretical support is required, especially in the part of the discussion of results, which allows the findings of this research to be contrasted with previous or related studies.
Author Response
Comment: In introductory part and even the initial part of the method, descriptions and statements are made that require theoretical or empirical support or perhaps a statistical source that supports some statements, for example, the behavior of the market, the economy, employment, etc.
Response: Appropriate references are provided in the revised version.
Comment: In general, it seems to me that greater theoretical support is required, especially in the part of the discussion of results, which allows the findings of this research to be contrasted with previous or related studies.
Response: In light of this comment the discussion of results is modified in the revised version.

Reviewer 5 Report
The authors present an interesting work for the region, correctly organized and with an important number of references.
The following suggestions are made for the paper, citing the lines where the observations are found to make it easier for the authors to locate them.
Lines 101-111: it is recommended to make smaller paragraphs of no more than 15 lines in order not to lose the idea.
-Line 108: reference is made to Table S1, review the rules of the journal normally cited as Table 1 without the accompanying letter S1, the same applies to the other tables cited in the paper.
Line 163: refers to Figure 1 and Line 177 refers to Figure S1, please review and correct.
Improve the quality of all figures so that they can be better visualized, following the recommendations of the journal for Figures.
Line 230: Figure 3 has connections, a recommendation for the authors would be to make connections that end in arrowheads to understand from which variable to which variable connects in one direction or bi-directionally, this allows better reading of the networks.
Line 234: there is an extra space between the word employment, average hours of work... Also in Line 302... between the word occupations, causal... in the same way in Line 218: between the word Nevertheless, for white-collar work... Line 378, at the beginning of the paragraph .... I recommend reviewing the whole paper to eliminate the extra spaces between words.
Additional:
Review the format when making citations of authors, the journal asks that as they appear these are placed between [1] when citing the first author or authors.... if there are more than two in the citation, for example, is placed [1-6]...review this section in the suggestions of the journal.
The work is interesting for the industry of the country where the study is applied, how complicated it can be to replicate it in other countries, and what restrictions may exist, it would be interesting to comment on it as part of the conclusions.
Author Response
Comment: The authors present an interesting work for the region, correctly organized and with an important number of references.
Response: We thank the reviewer.
The following suggestions are made for the paper, citing the lines where the observations are found to make it easier for the authors to locate them.
Comment: Lines 101-111: it is recommended to make smaller paragraphs of no more than 15 lines in order not to lose the idea.
Response: Necessary editing has been done.
Comment : Line 108: reference is made to Table S1, review the rules of the journal normally cited as Table 1 without the accompanying letter S1, the same applies to the other tables cited in the paper.
Response: Table S1 refers to the supplementary material that is submitted along with the paper.
Comment: Line 163: refers to Figure 1 and Line 177 refers to Figure S1, please review and correct.
Response: We have reviewed it. While figure 1 refers to the figure present in the main paper, figure S1 refers to the additional figure presented the supplementary material.
Comment: Improve the quality of all figures so that they can be better visualized, following the recommendations of the journal for Figures.
Response: Incorporated in revised version.
Comment: Line 230: Figure 3 has connections, a recommendation for the authors would be to make connections that end in arrowheads to understand from which variable to which variable connects in one direction or bi-directionally, this allows better reading of the networks.
Response: Since it is a network diagram, every tie is bidirectional. We have also clarified it now in the figure legend.
Comment: Line 234: there is an extra space between the word employment, average hours of work...
Response: Rectified in the revised version.
Comment: Also in Line 302... between the word occupations, causal... in the same way in Line 218: between the word Nevertheless, for white-collar work... Line 378, at the beginning of the paragraph .... I recommend reviewing the whole paper to eliminate the extra spaces between words.
Response: Editorial refinements have been made.
Additional Comment: Review the format when making citations of authors, the journal asks that as they appear these are placed between [1] when citing the first author or authors.... if there are more than two in the citation, for example, is placed [1-6]...review this section in the suggestions of the journal.
Response: Editorial refinements have been made.
Additional Comment: The work is interesting for the industry of the country where the study is applied, how complicated it can be to replicate it in other countries, and what restrictions may exist, it would be interesting to comment on it as part of the conclusions.
Response: We have added a few lines in the conclusion section in the revised version.
